# Food Security Interventions among Refugees around the Globe: A Scoping Review

**DOI:** 10.3390/nu14030522

**Published:** 2022-01-25

**Authors:** Christine Nisbet, Kassandra E. Lestrat, Hassan Vatanparast

**Affiliations:** 1Office of the Vice President Research, University of Saskatchewan, Saskatoon, SK S7N 0W9, Canada; christine.nisbet@usask.ca; 2College of Pharmacy and Nutrition, University of Saskatchewan, Saskatoon, SK S7N 5E5, Canada; kel547@mail.usask.ca; 3School of Public Health, University of Saskatchewan, Saskatoon, SK S7N 2Z4, Canada

**Keywords:** food security, food insecurity, refugees, intervention, displaced people, asylum seekers, scoping review

## Abstract

There are 26 million refugees globally, with as many as 80% facing food insecurity irrespective of location. Food insecurity results in malnutrition beginning at an early age and disproportionately affects certain groups such as women. Food security is a complex issue and must consider gender, policies, social and cultural contexts that refugees face. Our aim is to assess what is known about food security interventions in refugees and identify existing gaps in knowledge. This scoping review followed the guidelines set out in the PRISMA Extension for Scoping Reviews. We included all articles that discussed food security interventions in refugees published between 2010 and 2020. A total of 57 articles were eligible for this study with most interventions providing cash, vouchers, or food transfers; urban agriculture, gardening, animal husbandry, or foraging; nutrition education; and infant and young child feeding. Urban agriculture and nutrition education were more prevalent in destination countries. While urban agriculture was a focus of the FAO and cash/voucher interventions were implemented by the WFP, the level of collaboration between UN agencies was unclear. Food security was directly measured in 39% of studies, half of which used the UN’s Food Consumption Score, and the remainder using a variety of methods. As substantiated in the literature, gender considerations are vital to the success of food security interventions, and although studies include this in the planning process, few see gender considerations through to implementation. Including host communities in food security interventions improves the refugee–host relationship. Collaboration should be encouraged among aid organizations. To assess intervention efficacy, food security should be measured with a consistent tool. With the number of refugees in the world continuing to rise, further efforts are required to transition from acute aid to sustainability through livelihood strategies.

## 1. Introduction

There are 26 million refugees (“someone who is unable or unwilling to return to their country of origin owing to a well-founded fear of being persecuted for reasons of race, religion, nationality, membership of a particular social group, or political opinion” [1]) around the world (approximately 50% are children) along with another 45.7 million internally displaced people (“[those who] have not crossed a border to find safety. Unlike refugees, they are on the run at home” [2]) and 4.2 million asylum seekers (“someone whose request for sanctuary has yet to be processed” [3]) [4]. The top source countries of refugees as of 2020 include Syria, Venezuela, Afghanistan, South Sudan, and Myanmar [4]. While some refugees reside in camps, the vast majority live in makeshift cities and host communities in neighbouring countries, where rising tensions have been reported [5]. Some refugees are provided the opportunity to resettle in countries such as Canada or those in Europe and others are repatriated [6]. The instability of many countries around the world due to war, religious and cultural persecution, and environmental disasters continues to increase the numbers of people fleeing their homes every day.

Recent conflicts around the globe are creating larger numbers of refugees for more prolonged periods of time. In desperation, refugees pay to board unsafe, overloaded water vessels. Many do not make it across with the number of dead and missing at its highest of more than 5000 in 2016 [7]. Families are forced to separate, people are met with long wait times trying to enter refugee camps in neighbouring countries, and some countries close their borders forcing refugees to search for asylum elsewhere. The protracted nature of crises such as that of Afghanistan and Syria strains host countries and aid agencies, stretching resources thin and impacting health care, food security, and livelihoods.

The United Nations Refugee Agency (UNHCR) reports that 80% of the world’s displaced people are in locations suffering from acute food insecurity and malnutrition [8]. The COVID-19 pandemic continues to exacerbate the situation. Food security exists when all people at all times have access to safe and nutritious food appropriate for culture and lifestyle [9]. Food security must be examined across four pillars: physical availability of food, economic and physical access to food, food utilization, and stability over time [10]. The right to food and food security cannot exist without addressing the restrictive laws and policies refugees face in many countries such as those around employment and freedom of movement [11], yet food security remains a complex issue. 

Particular consideration is needed for the most vulnerable, including women and girls; children; lesbian, gay, bisexual, transgender, queer and/or questioning, intersex, asexual, two spirit, and others (LGBTQIA2S+); the elderly; and persons with disabilities [12,13]. Women and children are particularly at risk of violence, sexual exploitation, and abuse as families are often separated during migration and refugees are forced to seek help from smugglers and others who take advantage of them, and refugee camps have high population densities with limited services. LGBTQIA2S+ populations are discriminated against, harassed, abused, and murdered, particularly in countries with anti-LGBTQIA2S+ legislation [14]. The elderly and persons with disabilities face barriers when it comes to accessing resources—such as water if collection points are far from their shelter—healthcare, and other services [15]. Gender roles are important in terms of household finances and food security. All over the world, women have a slightly higher prevalence of food insecurity compared to men [16]. Women are likely to prioritize food needs of spouses and children while compromising their own [17,18]. Culture is another important consideration because a lack of culturally available foods can destabilize cultural identity, affecting both physical and mental health [19]. Food insecurity results in a double burden of disease where malnutrition in childhood is followed by early establishment of chronic diseases later in life. The 2020 Global Nutrition Report indicates that 149 million children less than five years of age are stunted, 50 million are wasted, and 40 million are overweight [20]. Malnutrition is very common in refugee children [21]. Information is available on women and children; however, LGBTQIA2S+, the elderly, and those with disabilities are often overlooked in the design and implementation of humanitarian aid [14,15,22].

Some people live their whole lives in refugee camps with little hope of an autonomous future, yet they do what they can with their limited resources to survive. In camps, refugees are reliant upon aid, provided rations, vouchers, or cash for food. In makeshift cities, they can remain isolated from the host community where significant tensions exist [5,23]. Supplemental and therapeutic feeding centres are common for infant and young child feeding, target both refugees and host communities, and have proven to be successful in addressing malnutrition [24]. A vital strategy is to work with host countries to provide refugees with documentation to allow them the same rights as other citizens so they can access basic necessities like education, healthcare, and employment [25]. Such approaches improve self-reliance and mental health and provide training opportunities for refugees to build the gap between market demand and refugee skills, considering gender and other social and cultural contexts [26].

In resettlement countries, food security remains an issue. Migrants find cultural foods expensive, hard to obtain, and although people often have cooking skills, the unfamiliarity of new foods and how to prepare them pose challenges [19]. Refugees are provided aid for a short period of time, but many barriers such as language and lack of recognition of education from their home country makes it difficult to land jobs that pay well. For example, preliminary data indicates that 70% of Syrian refugees in Canada experience food insecurity [27].

Many countries around the world are welcoming refugees and donating money towards helping those in need. For example, United Nations Agencies and nongovernmental organization (NGO) partners pledged $5.5 billion USD to assist Syrian refugees in 2020 [28]. With the numbers of refugees continuing to rise year after year, we need to review how we are helping these vulnerable people. Therefore, it is important to assess the types of food security interventions and identify the gaps in research to inform future programming to maximize efficiency of resources and help the largest number of people by the greatest extent possible. 

### Objectives

The objective of this scoping review is to assess what is known about food security interventions in refugees and identify existing gaps in knowledge.

Although our ultimate interest is refugees, interventions aimed towards other populations such as asylum seekers and displaced persons would be similar and so were also included. Interventions included formal interventions from research and humanitarian aid agencies such as cash and food transfers, food vouchers, urban agriculture, community gardens and kitchens. We are interested in knowing what interventions are most successful for refugees. We know that Community Based Participatory Research should be prioritized for successful interventions, placing the population of interest at the core, and engaging them throughout the entire research process. Therefore, we also included informal interventions implemented by refugees themselves such as the development of informal economies (“the diversified set of economic activities, enterprises, jobs, and workers that are not regulated or protected by the state...[including] wage employment in unprotected jobs” [29]). Although we are interested in which interventions are most successful, we also need to know what has been attempted with minimal to no success. Therefore, instead of only including successful interventions, we included all interventions. Refugee food security is a global issue, thus our review includes interventions from all countries, keeping in mind that different types of interventions will be observed according to where the country is along the migration process, from point of entry countries to transit countries, to final destination countries.

## 2. Methods

This scoping review followed the guidelines set out in the PRISMA Extension for Scoping Reviews (PRISMA-ScR): Checklist and Explanation (2018) article [30].

### 2.1. Eligibility Criteria

Inclusion criteria for this scoping review included any article that discussed a food security intervention in refugees. Articles were excluded if they were published prior to 2010, were not available in the English language, were not about food security interventions in refugees, or were exploratory studies, protocol or framework papers, conference abstracts, or review articles. For articles published by UN agencies, only those with an accompanying evaluation were included to incorporate a measure of effectiveness of interventions.

### 2.2. Information Sources and Selection

The search was executed on 29 June 2020 in Ovid MEDLINE, Global Health, Public Health Databases, SCOPUS, and CABI Abstracts Global Health (from Web of Science). The search strategies were developed in consultation with the research team and a librarian experienced in scoping reviews. A sample search strategy from Ovid MEDLINE can be found in Appendix A: Sample search strategy. Search results were exported to EndNote X9 3.3 and duplicates removed [31]. Articles published from 2010 to 2020 were scanned in the Journal of Refugee Studies, the Journal of Immigration and Refugee Studies, and the Emergency Nutrition Network. The reference lists of all included studies were scanned for articles published from 2010 to 2020 that met the eligibility criteria. Grey literature was also included by scanning United Nations (UN) websites including the UNHCR, the Food and Agriculture Organization of the UN (FAO), the World Food Programme of the UN (WFP), and the World Health Organization. The titles and abstracts were scanned for eligibility criteria by authors CNN and KEL while any disagreements were discussed amongst all authors (CNN, KEL, and HAV) until consensus was reached.

### 2.3. Data Charting Process and Data Items

A form was developed in Microsoft Excel to extract all necessary details from the included articles: study location, study design (sample sizes at the household/family level vs. individual level and in the intervention vs. evaluation including pre and post), food security measurement tool, participants (age and gender), whether or not the intervention considers gender and any other at-risk groups, outcomes/important results, and limitations. Authors CNN and KEL charted the data and updated the form in an iterative process.

### 2.4. Synthesis of Results

Results are presented using a series of tables and figures to best depict the different results.

## 3. Results

### Selection and Characteristics of Sources of Evidence

The removal of duplicates left a total of 4134 citations from electronic databases, journal scans, and reference list searches. Scanning titles and abstracts based on the eligibility criteria outlined above resulted in the exclusion of 4001 articles. We then went through 133 full text articles, whereby another 76 were excluded for not being about refugees or not distinguishing refugees from other population groups (e.g., immigrants), not including an intervention (cross-sectional, exploratory, simulation), or being a review, opinion, or policy. Therefore, a total of 57 articles were eligible for this study (Figure 1).

Table 1 is organized by the first author’s last name and provides details on the characteristics of all included articles including aim, study design, and outcomes. We examined articles by location and found that 32% targeted refugee camps and/or settlements, 19% were outside camps, 26% were both inside and outside camps, one article did not specify, and 21% were in destination countries (Figure 2). We also found that 67% of interventions targeted refugees only, while 33% targeted both refugees and host communities. Only 47% of the articles indicated a consideration for gender when designing and implementing the interventions (i.e., programs were targeted specifically to women and/or women were prioritized by being provided e-transfers to manage household expenses or given roles to oversee food distribution). Few studies mentioned other at-risk populations such as children not covered by IYCF programs, the elderly and persons with disabilities. None of the studies mentioned LGBTQIA2S+. We found that 26% used a mixture of cash, vouchers, or food transfers for the intervention while another 11% were cash only interventions and 2% were voucher only. We also found that 28% of interventions were on urban agriculture, gardening, animal husbandry, or foraging; 12% on a combination of nutrition education type interventions; 12% on infant and young child feeding; 4% focused solely on school-based nutrition; 2% on community kitchens specifically; 2% on food safety and energy; and 2% on informal economy/trading (Figure 3). Table 2 is organized by location and provides details on the emerging themes from our results including location, target population, intervention type, consideration for gender, and food security measurement tool. Results indicated that 55% of interventions in nondestination countries were led by UN agencies of which 64% involved cash and/or vouchers; 20% used urban agriculture, gardening, and animal husbandry; and 16% were on infant and young child feeding and pregnancy. For destination countries, one was in Canada, one in Germany, two in Australia, and eight in the USA (Figure 4). All seven of the nutrition education interventions took place in destination countries, representing 58% of the destination country interventions. The other interventions in destination countries involved urban agriculture (25%), infant and young child feeding and pregnancy (8%), and cash (8%).

Food security was directly measured in 39% of studies. While the remainder addressed food security with interventions such as urban agriculture, infant and young child feeding, and nutrition education programs, they did not include direct assessment. While 52% of studies that measured food security used the United Nations Food Consumption Score alone or in addition to the accompanying Diet Diversity Score or Coping Strategies Index, the other 48% each used different methods to measure food security. These methods used one question or multiple questions to assess food security status. Although many seem to be based on the FAO, the WFP, or the United States Department of Agriculture (USDA) methods—the three most well-known validated questionnaires in developing and developed countries—there is no indication that they were validated questionnaires.

## 4. Discussion

In this review, we evaluated food security interventions in refugees and existing gaps in knowledge. Overall, 57 studies met the inclusion criteria, mainly in the area of refugee crisis. Consistently high levels of food insecurity among refugees indicate a need for one standard tool to measure food security across locations to improve understanding around food security in different contexts and help determine best practices and policies. This review has discovered multiple gaps in research leading to limited knowledge of the efficacy of interventions in different refugee settings. 

### 4.1. Summary of Evidence 

#### 4.1.1. Intervention Types across Geographic Locations 

##### Areas of Refugee Crisis 

Most studies in areas of refugee crisis such as the Middle East and Southeast Africa report on interventions that include a mixture of cash, vouchers, and food transfers (Figure 3). Substantive literature exists on types of interventions, providing evidence for cash-based transfers as opposed to vouchers or food rations as cash provides choice, flexibility, sense of dignity, and empowerment [23,32,33]. However, in areas where markets are not developed, such as newly established refugee camps, rations seem to be the most beneficial until informal and/or formal economies are established, and markets stabilize. When providing assistance, it is important to consider gender, the inclusion of host communities in the interventions, and the accompaniment of livelihood strategies.

When examining intervention types by UN agencies, we observed that urban agriculture was a focus of the FAO, and cash/voucher interventions were implemented by the WFP; however, there was not much mention of these two agencies working together to combine efforts. The FAO aims to achieve food security for all, the mandate of the UNHCR is to provide international protection to refugees and other persons of concern, and the role of the WFP is to use food aid to support economic and social development, meet food needs in emergency and protracted situations, and promote food security based on FAO recommendations [87,88]. Despite documents such as the 2011 Memorandum of Understanding between UNHCR and WFP being in place, details of these collaborations are lacking, and evaluations of UN agency programs recommend collaboration [81,88]. For example, a 2016 evaluation of WFP programs in Liberia indicated that UNICEF and FAO are listed as partners in the project document, yet no evidence of this collaboration could be found by the evaluation team in any other documentation. Inter-agency action-oriented collaboration could maximize resources, streamline services, and allow the development of successful plans for a transition from cash assistance to livelihood strategies and thus programmatic sustainability. 

Based on recommendations from UN agency impact evaluations, in July 2020, the UNHCR and the WFP announced the launch of the “Joint Strategy for Enhancing Self-Reliance in Food Security and Nutrition in Protracted Refugee Situations” [89]. They will assess the refugee situation together, investigate the vulnerabilities, capacities and opportunities together based on their assessment, and set goals to improve self-reliance and livelihoods [89]. They will also evaluate their progress on self-reliance in food security together [89]. The new strategy has two main objectives that focus on empowering refugees and creating a supportive environment by engaging the local government and host communities [89]. Although the new joint strategy seems promising and focuses on empowering refugees by engaging all stakeholders, to our knowledge, there is no evidence to evaluate its effectiveness. 

In areas of refugee crisis, when host communities are not involved in interventions, it creates feelings of hostility towards refugees as host communities feel like refugees are being helped above their own most vulnerable. The refugee–host relationship can also be affected by country policies which limit the rights of refugees limiting freedom of movement, access to work visas, ownership of land, and more, which is beyond the scope of this review. Including host communities when targeting households for food assistance improves the refugee–host relationship [46]. 

Livelihood strategies are important to improve sustainability of the aid provided and assist refugees in becoming self-sufficient, particularly when aid is often reduced [11]. It is of note to mention that not all interventions are purposeful, and some are instigated by refugees themselves in the form of establishing informal economies and trading in and around refugee camps [90]. It is beneficial to take note of these interventions as well because we can learn from the entrepreneurial activities of refugees when planning interventions as it is indicative of what refugees need. By providing more livelihood opportunities with the support of humanitarian aid agencies, it may be possible to improve refugee self-reliance, empowerment, and gender equity [46,47,60,76]. 

A considerable amount of evidence is focused on Palestinian and Syrian refugees in Lebanon and Jordan [5,41,43,48,58]. While most interventions in refugee crisis areas are focused on cash, vouchers, and food transfers, studies in Lebanon reported more sustainable programs such as school-based nutrition, community kitchens and urban agriculture, which are in the line of main interventions in developed destination countries [41,43,45,48,58]. 

##### Destination Countries

Refugees are a vulnerable population that suffer unique challenges that often affect their food security status even after entering destination countries. Our results showed few studies are being conducted on refugee food security interventions in developed destination countries despite similar levels of food insecurity between refugees in destination and nondestination countries [6,91]. For example, a Canadian study by Lane et al. (2019) reported that 50% of refugee households (from various countries of origin) were food insecure [91]. Similarly, 50% of Syrian refugees in Lebanon have been found to be food insecure [6]. It is also common to see studies in destination countries (e.g., Canadian Community Health Survey in Canada) grouping refugees with immigrants and excluding participants who cannot speak the country’s official languages, which portrays an inaccurate and underestimated image of refugee food security issues [92]. Only 17% of refugee food security intervention studies in destination countries measured food security status.

There is a significant difference in the types of interventions in developed destination countries focusing mainly on urban agriculture, gardening, animal husbandry, and foraging, and other nutrition programming such as nutrition education (Figure 3). In destination countries such as Canada, refugees are covered by direct cash support and housing programs in the first year of arrival [93]. Afterwards, based on their situation, they could be eligible for regular social assistance programs. An abrupt cessation to federal government aid may explain the high prevalence of food insecurity among refugees in destination countries a year after arrival [27].

#### 4.1.2. Considerations for the Most Vulnerable

Gender is an important consideration when developing food security interventions. In many cultures, women are often in charge of food preparation for the household. We know that women/mothers are more likely to cut back their intake and portion sizes so that other families, particularly children, can have enough to eat [17]. Women are more likely to be food insecure and women and girls are at greater risk of gender-based violence [50,82,85,94]. Although many UN agency interventions included gender considerations in the intervention plan (e.g., planned to target women as beneficiaries of cash/food transfers), evaluations showed that these considerations are lacking during implementation [81,85]. Evaluations often indicated a need for more security, oversight, monitoring, and evaluation in camp settings [81,84]. Equitable gender considerations can be difficult because many countries still lack women’s rights and their policies and social norms may prevent women from seeking employment outside the home, and other genders are not considered due to discrimination and oppressive laws [14,32,94]. Few studies mentioned other at-risk populations such as children not covered by IYCF programs, the elderly, LGBTQIA2S+, and persons with disabilities, and research shows that these people are often overlooked in the design and implementation of humanitarian aid, indicating a need to amend interventions to assist these at-risk groups [14,15]. Although not all interventions can affect policy change, it is important to work with governments to find ways to assist the most vulnerable.

#### 4.1.3. Assessing Food Security

Our review showed that less than half of the studies that aimed to address food security issues actually measured food security, and those that did used a variety of different tools with only some being validated. The most common tool used to measure food security was the UN’s Food Consumption Score used in 52% of the studies that measured food security, while all other tools were only used in one study each. A wide range of food security topics makes it difficult to assess the efficacy of interventions. A consistent tool that is validated in different languages is needed to accurately compare food security across locations and contexts, differentiating between adult and child food security and providing a more complete picture of food security issues in households, which would allow more targeted interventions. The WFP is evaluating the food security status of refugees in areas of refugee crises using the Consolidated Approach to Reporting Indicators of Food Security (CARI) [95]. This comprehensive tool incorporates the Food Consumption Score, economic capacity, and livelihood coping strategies, which has been widely accepted and is a good measure of food security [95]. The Household Food Security Survey Module (HFSSM) is a questionnaire containing 18 questions that assess income-related food security status at household, adult, and child levels [96]. The HFSSM has been validated and used in more than 19 languages in different countries, particularly developed destination countries [96]. The ability of the HFSSM to assess food security in households, adults, and children makes it a proper candidate as a standard tool that fills the gap in our ability to universally assess the efficacy of food security interventions in different settings. Destination countries such as Canada and the USA are using the HFSSM regularly in their nutrition and health national surveys [96]. Therefore, using either tool or a combination as a standard food security assessment tool will allow the comparison of food security status of refugees with host countries to identify the gaps and disparities. 

### 4.2. Knowledge Gaps and Research Recommendations

A considerable number of studies in areas of refugee crisis evaluated the short-term interventions of international agencies individually [23,81,82]. There is a lack of evidence as to whether international agencies are working together on interventions they support collectively and, if so, how effective those initiatives are compared to interventions implemented by one agency alone. Further, it is not clear the extent to which international agencies work with local governments or NGOs on the sustainability of interventions that is necessary to empower refugees, enable them to be self-sufficient, improve their food security status, and contribute to local economies. 

Research has shown that beneficiaries prefer cash to vouchers and rations and that cash often results in better outcomes compared to other modalities [32,33,39,40,56]. The lack of direct food security measures in many studies, along with insufficient methodologies (e.g., measures only in one time point, lack of food security measures, lack of control group), prevented an assessment of any improvement correlated with the intervention itself. The lack of a consistent tool used to measure food security prevents any comparison across studies, which goes beyond the scope of this review. Similarly, limited studies on cash, vouchers, and/or rations measured food security and considered gender in their implementation. Of those that did, none compared food security results across genders.

Grey literature indicates the role of community-based organizations and host communities in supporting and empowering refugees, particularly in destination countries [60,61,77]. Such organizations conduct interventions without proper pre- and post-evaluations, leading to lack of evidence on the impact and effectiveness of such initiatives. There is a need to identify, evaluate, and document best practices aimed to improve the food security status of refugees. 

Although international agencies have clear policies and work plans with regards to food security in areas of refugee crisis, to our knowledge no study has evaluated the policies by local governments in areas of refugee crisis as well as destination countries [88,97,98]. Such studies will assist in identifying effective policies that aim to improve food security status of refugees while empowering them as new members of the host community. 

Short-term interventions are necessary to alleviate hunger and other short-term effects of food insecurity among refugees. However, many protracted refugees continue to live in unstable situations in host countries, which can impact their food security status. There is limited information surrounding food security interventions in protracted crises, likely due to limited resources and international aid agencies focusing efforts on acute crises. Thus, further efforts are required to address sustainability issues when it comes to food security interventions. 

### 4.3. Strengths and Limitations 

To our knowledge, this is the first study to use a systematic approach using the PRISMA guidelines to identify and evaluate the selected literature on refugee food security interventions. The main strength of our study is the systematic method of setting eligibility criteria, identifying the literature, and detecting the gaps in research. The categorization of types of interventions and geo-mapping according to geographic locations is another strength of our study that provides insight into the distribution of the types of interventions across the globe. 

Regarding limitations, we only included interventions reported in the English language as indicated in the inclusion criteria. Therefore, we were unable to identify and include reports available in different local languages. The variation in the tools used to assess food security and methods of evaluation limited us from having an overall picture of food insecurity status before the evaluation of the effectiveness of interventions. 

## 5. Conclusions

Refugee crisis is on the rise due to climate change, war and other political and societal conflicts. Humanitarian agencies continually provide assistance and evaluate their interventions in areas of refugee crisis. The resultant evidence has provided substantive information on when to use certain types of interventions, such as cash when markets are stable and the importance of incorporating livelihood strategies to transition to a sustainable level of aid and help refugees become self-sufficient and active members of their communities. In destination countries, the types of interventions are more towards capacity building and education. Considering numerous existing interventions, the rate of food insecurity is still very high among refugees. In addition, due to lack of a proper and universal approach for evaluation, the efficacy of interventions is not clear. Further efforts are necessary to work with governments to affect policy change to advocate for the rights of marginalized populations such as children, seniors, women, LGBTQIA2S+, persons with disabilities, and minority groups. It is also vital to engage host communities and NGOs to create a welcoming culture that benefits both refugees and host communities. Finally, researchers should adopt a standard feasible food security assessment tool which is needed to assess the effectiveness of interventions across locations and countries to develop best practices based on comparative results.

## Figures and Tables

**Figure 1 nutrients-14-00522-f001:**
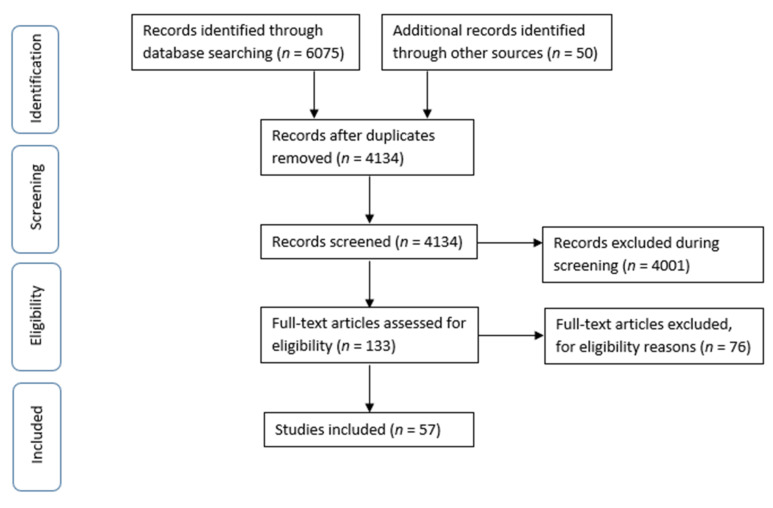
PRISMA Flow Diagram.

**Figure 2 nutrients-14-00522-f002:**
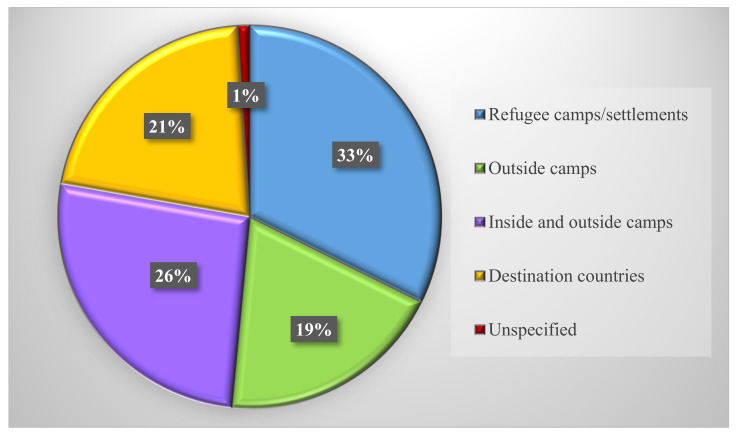
Percent of articles by location.

**Figure 3 nutrients-14-00522-f003:**
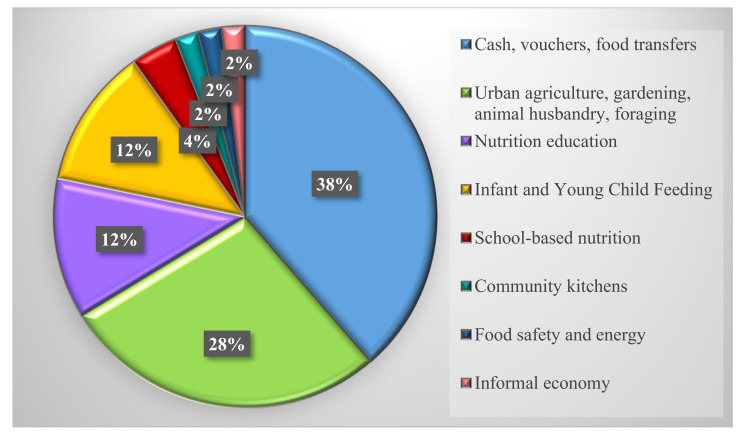
Percent of articles by intervention type.

**Figure 4 nutrients-14-00522-f004:**
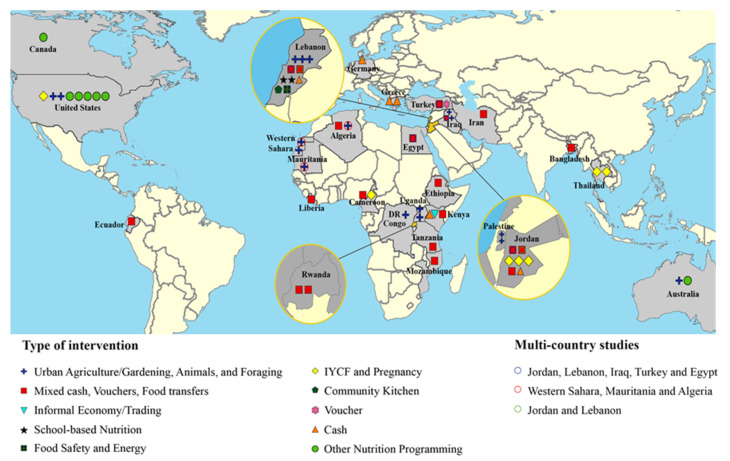
Intervention types according to country.

**Table 1 nutrients-14-00522-t001:** Characteristics of Sources of Evidence.

First Author, Year; Location	Aim	Intervention	Participants	Results
Abu Hamad B, 2017; Jordan, Amman, Irbid, Mafraq and Zarqa [32]	Find effects of UN Refugee Agency (UNHCR) cash, UNICEF Child Cash Grant (CCG) on beneficiaries’ lives: spending patterns, well-being; efficiency, effectiveness, accountability of cash provided; complementarity of and gaps in programming for most vulnerable.	6 groups: 1. cash, CCG, and full-value vouchers; 2. cash, CCG, and half-value vouchers; 3. cash and full-value vouchers; 4. cash and half-value vouchers; 5. full-value vouchers only; 6. half-value vouchers only. Eval-random selection, mixed methods: lit review, quantitative surveys, focus groups, key informant interviews, case studies.	2114 household surveys: 627 cash, CCG, full-value vouchers; 418 cash, CCG, half-value vouchers; 165 cash, full-value vouchers; 42 cash, half-value vouchers; 251 full-value vouchers; 611 half-value vouchers. Purposive sampling for qualitative interviews-432 adults/children had different types of aid.	Positive perceptions of cash. Cash = avoid coping strategies (e.g., eat less, remove kids from school). Borrowing money ↓ 79% to 26%. 90% said cash helped pay rent, 40% moved to better housing. 27% of all types of cash could not pay rent vs. 52% vouchers only. Cash, CCG less likely to have food shortages, forgo meat, eggs, dairy; more likely to have acceptable food security (90% vs. 82%).
Alloush M, 2017; Rwanda, Kigeme, Nyabiheke, and Giheme camps [33]	Characterize demographics and income generating activities.	Cash camps: monthly transfers (m-VISA) on cell phones to gain cash, purchase goods/services. In-kind camps: monthly basket of maize, beans, oil, salt. Surveys: how camp economies interact with host-country economies; local economic impacts of in-kind vs. cash. Kigeme = in-kind; Nyabiheke = cash; Giheme = cash.	Random sample of households. Congolese refugees: 155–224 per each of 3 camps; host-country: 162–243 in economically relevant sectors 10 km of camps. Additional businesses: 15–23 refugees in each camp and 63–100 hosts at main commercial sites within 10 km radii.	>80% of refugees sold food aid to purchase food, nonfood items. Refugees in cash camps better off than in-kind. Food security at Kigeme 14%, Nyabiheke 39%, Gihembe 60%. Despite poor circumstances, economies form in camps-exchange of goods, services within/between camps and host economies.
Alsamman S, 2014; Jordan, Za’atari camp [34]	Establish 3 caravans serving as mother-baby friendly space.	Promote caravans as safe for breastfeeding (bf)-privacy, support. Topics: nutrition for pregnant and lactating women, importance of bf, complementary feeding, feeding during illness	Pregnant women and mothers with children <5 years. Reached 15,600 mothers >18 months	Emphasized exclusive bf, time for complementary feeding. Identified bf difficulties, provided aid. ↑ awareness of risks of infant formula. Fortified food for 6–23 months distributed monthly; no 4th cycle-inadequate funds.
Aste N, 2017; Lebanon, refugees in camps and informal settlements [35]	Improve food security (food utilization) by testing energy technologies, mainly related to cooking, food preservation.	Case A: Electricity previously few hrs/day did not reach all households, unofficial connections = unsafe exposure of electricity. New system with security, safety. Case B: previous lack of food preservation capacity, illegal connection to grid, use of private generators for lighting. Added system for each family, charge controller, fuse for system and user safety = electricity for fridge, lights.	Case A: Converted shopping mall with 134 families (670 people). Case B: 82 refugees in rural Lebanon living in shelters and uncompleted buildings.	Case A: Diet diversity ↓ due to ↓ value of vouchers. Diet diversity of women = ↑ trend-fridges stored food longer, safely. Fridges ↓ expenses. Fridge internal temp not as low as expected, preserve water, bread. Case B: fridges preserve some food for limited time due to ↓ food availability, unfamiliarity of refrigeration. Food still perceived safer, healthier.
Battistin F, 2018; Lebanon, North, Beirut, Mt. Lebanon, Bekaa, and South [36]	Measure impact of Multipurpose Cash Assistance Programme (MCA) delivered by Lebanon Cash Consortium at 6-month midline on several proxies of physical and material wellbeing: food security, health, hygiene and housing.	Quasi-experimental, Regression Discontinuity Design; to compare outcomes of households that received cash vs. those who did not (non-MCA received vouchers).	20,000 of 25,000 refugees eligible for MCA were assisted due to lack of funds. Eval: compared 261 MCA and 247 non-MCA households; most male-headed (>75%).	MCA ↑ consumption of food, gas for cooking. Food expenditures 33% ↑ for MCA vs. non-MCA. Food security indicators not sig impacted by MCA, but were for non-MCA. No impact on food-related coping strategies; both coped similarly. MCA effective to address barriers where markets functioning, flexible to demand.
Betts A, 2020; Kenya, Kalobeyei settlement and Kakuma camp [37]	Provide self-reliance to refugees, greater refugee–host interaction through development of Kalobeyei settlement, planned for refugees living on one side, hosts on the other, with shared markets, schools, hospitals in the middle.	Bamba Chakula programme: monthly mobile cash transfers for food at registered shops. Kalobeyei: cash, corn-soya fortified powder. Kakuma: cash, food basket. Agriculture promotion programme encouraged self-reliance, included kitchen gardens, community plots. Eval: quantitative survey to compare self-reliance of recent arrivals, focus-groups, semi-structured interviews.	2560 surveys, 15 focus groups, >40 semi-structured interviews with refugees primarily from South Sudan, smaller numbers from Ethiopia Burundi, DR Congo, Uganda, Sudan, Somalia; nonrefugee stakeholders; gov officials; host community.	In Kalobeyei 36% of South Sudanese had kitchen gardens vs. Kakuma 24%. Barriers: lack of water 90%, seeds 66%, equipment 29%, soil quality 21%. Rights to work restricted in Kalobeyei. 10% earned money, still low income. Public services limited in both camps. Acceptable diet diversity in Kalobeyei 66–76%, Kakuma 58%. Food insecurity in Kakuma 93%, Kalobeyei 78–90%.
Bloom JD, 2018; USA, North Carolina [38]	↓ social isolation, ↑ access to resources, adapt more generally to USA food systems by facilitating immigrant, refugee communities’ ability to apply healthy traditions in a new context.	Asset mapping workshop with women’s committee. Worked with 2 communities, partner orgs 1 y to develop projects, evaluate. Mosque home garden project: attend class, provided materials, supplies. Karen: connected with local nonprofit incubator farm, provided training, tools, seeds.	Evaluative interviews with 6 women from women’s committee at local mosque, out of 27 total participants. 6 interviews with 7 of 8 participants from Karen community.	Most mosque participants did not produce enough veg to improve intake. School garden food brought to mosque, given out free. Karen participants decreased store purchases, improved access to healthy, traditional food, shared with 3–30 refugee families.
Boston Consulting Group, 2017; Jordan (Amman, Balqa, Irbid, Mafraq) and Lebanon (Beirut, Bekaa, Mt. Lebanon, North, and South regions) [39]	Compare impact of WFP assistance-delivery modality, cash, food vouchers on food security, other basic needs of refugees in host communities; cost-effectiveness.	RCT, 3 groups: voucher, cash, choice (e-voucher, cash or mixed). In Lebanon data collected at 2 post-distribution monitoring (PDM) points over 5 months. In Jordan, 3 PDM points over 8 months. Each PDM: quantitative survey, focus groups. Other indicators: bank, retail transactions; food prices.	3123 households. Jordan 1848 households, ~300 cases per vulnerability level in each of 3 treatment groups. Lebanon 1275 households (425 per group).	>75% preferred cash: ↑ purchasing power, flexibility, capacity to manage cash, dignity, empowerment. Food security better or = with cash vs. vouchers. Diet diversity optimal in 70–72% cash, 62–67% voucher. Cash = more nutritious food. Coping strategies, spending patterns equal.
de Bruin N, 2019; Tanzania, Nyarugusu Refugee Camp (Burundi and Congolese) and Tanzanian citizens active in the common market or from 3 villages (Mbwana, Ngasa, and Moshi) [40]	Examine effects of cash-based transfer program in Nyarugusu refugee camp.	Single case studies used purposeful sampling considering gender, age, nationality, role, expected knowledge. Data collected by observation, qualitative semi-structured interviews.	50 interviews: 27 refugees from Nyarugusu Refugee Camp (cash and food beneficiaries); 16 Tanzanian citizens (village leaders, farmers, businesspeople); 7 other stakeholders.	Preference for cash-improved choices, relationships with hosts. 75% thought market prices high, ↑: ↓ purchases, ↑ hunger. Village leaders: refugees ↑ economy. Shop owners/farmers: stronger infrastructure, more crime. Stakeholders: food supply in host community cannot meet camp needs, ↑ demand from cash = ↑ prices, undermining food security of poorest.
Dehnavi S, 2019; Lebanon, refugees and host communities [41]	Alleviate hunger and underweight among participants by improving food security and economic resilience through improved food availability via home gardens.	A closed-ended survey evaluated participant satisfaction with the project, challenges and measures taken to overcome them, and demographics and gardens characteristics. Total population sampling.	73 (72 female) households provided planting kits; 71% Syrian refugees, 29% Lebanese; 67% aged 30–45 years. 41 participants took part in the survey.	Minorly alleviated underweight or hunger. Crop production, cultivation low; 67% ↑ availability of fresh food, fruit/veg intake; diet diversity. 29% satisfied: 61% lacked water, 56% ↓ production than expected, 53% limited inputs, 17% unable to produce types of plants wanted, 10% intending to sell products did not-low production.
Dunlop K, 2018; Greece, all [23]	Enhance the wellbeing of Persons of Concern in Greece through access to protection-based and multi-sectorial humanitarian assistance.	Mixed methods. Eval outcomes: persons of concern can meet basic needs safely with dignity, choice; relationships with host communities improve. Eval examines negative coping strategies, links to local Greek economy through market cash injections via household surveys, focus groups, key informant interviews.	63,051 people received €6.3 M. Quantitative data examined 400 (44% Syrian, 25% Iraqi, 16% Afghani, 9% Iranian, 6% other) household surveys. Qualitative data from 6 focus groups, 21 key informant interviews. 327 male, 73 female due to men more often listed as head of household/card holder.	Eval: most highly reported areas of spending: unmet needs at baseline = success of multipurpose cash grants. 71% felt cash partially met needs. Most frequently unmet needs: clothing 69%, cigarettes 29%, debt repayment 23%. Most cash spent on food (77%), ↑ with ↑ households, Syrians, Iraqis. Coping strategies: eat less preferred/expensive food 70%, ↓ meals/portions 45%.
Eggert LK 2015; USA, The Shenandoah Valley region of Virginia [42]	Combat physical and mental health conditions that accompany migration by developing a community coalition to implement a community garden with apartment-dwelling refugees.	Recruitment: community liaison, snowballing. Coalition: fidelity in process, satisfaction. Garden: fidelity to construction, participation, satisfaction. Seeds provided, gardeners contributed tools, attended planting/training day. Gardens assessed through season, advice available.	5 gardeners began the season, 4 remained (1 moved).	More veggies consumed, most donated some of their garden, some liked not having to go to the store, all wanted a larger plot of land to garden. Two refugee residents observing community garden plots expressed interest in larger-scale urban farming.
El Harake MD, 2018; Lebanon, cities of Majdal Anjar, Saadneyil, and Bar Elias in the Bekaa valley [43]	Evaluate a 6-month pilot school-based nutrition intervention on changes in diet knowledge, attitudes, behaviours of Syrian refugee children in informal primary schools in rural Lebanon; explore the effect of the intervention on diet intake, nutrition status of children.	Quasi-experimental design. 2 intervention schools: health and nutrition education bi-weekly, nutritious snacks. Control school: usual curriculum, standard snack. Interviews with children, mothers: household sociodemographics, diet knowledge, attitude, child behaviour, anthropometric measures, diet intake.	Data collected at baseline from 296 Syrian refugee students 6–14 years (grades 4–6). Data at baseline and follow up available for 203 children. Sample size reduced to 183 due to clustering. At baseline, mean age of children was 11 years, 51% female.	Baseline: 79% severely food insecure, 3% food secure. Greater change in knowledge, body mass index-for-age (z score) and height-for-age (z scores) in intervention vs. Control. Compared to control, intervention children had on average sig ↑ mean changes in daily intakes: kcal, dietary fiber, protein, saturated fat, vit K, zinc, calcium, magnesium.
Fander G, 2014; Jordan, 6 northern governates (Amman, Zarqa, Mafraq-including Za’atari refugee camp, Irbid, Jerash and Ajloun) [44]	Protect children <5 years and pregnant and lactating women (PLW) by screening for malnutrition and educating caregivers about infant and young children feeding practices.	Pre-intervention assessment: breastfeeding misconceptions. Project: education on exclusive breastfeeding, correct/timely introduction of complementary foods via clinics with nutrition officer or similar; support for mothers willing to re-lactate; Super Cereal Plus supplement to treat moderate acute malnutrition in kids <5 years, PLW.	Over 10 months, 4690 pregnant/lactating women received education and 919 mothers engaged in breastfeeding counselling.	Increase in breastfeeding knowledge, but not in breastfeeding practice. Out of 46,383 children screened, 69 had severe acute malnutrition, 124 had moderate acute malnutrition. Out of 10,088 PLW screened, 457 were acutely malnourished.
Food and Agriculture Organization of the United Nations, 2016; Lebanon, Akkar, Tripoli, and the Bekaa regions [45]	Promote diversified and quality food for vulnerable landless households through micro-gardens.	6 micro-garden structures tested, as well as one method with no structure (plastic crates distributed for use as planter boxes). Initial training: technical support, follow-up of weekly site visits. Successes and/or failures of each method recorded, analyzed.	170 direct beneficiaries (76 vulnerable Lebanese and 94 displaced Syrians).	Vertical planting had lowest success, simplest structures best. Plastic crates cheapest, easiest, most successful, more easily accepted. Other factors impacted success: space, pest-resistant seed, reliable water supply, extreme weather protection. Micro-gardens ↑ quality of life, not a replacement for agriculture. Learnings: restrict to cooler seasons, use more pest-resistant leafy veg, herbs.
Food and Agriculture Organization of the United Nations, 2018; Uganda, refugee settlements in the northern and mid-western regions [46]	Improve food, nutrition, income security of refugees, host communities.	Planting materials and inputs for small-scale veg, staple food, poultry production, preservation techniques provided with construction, use of energy saving stoves, training in entrepreneurship and animal husbandry.	8000 households of most vulnerable refugee, host community households.	More diversified income sources; ↑ food security, diets; stronger livelihoods of vulnerable refugee/host communities. Beneficiaries learned skills, ↑ knowledge, ameliorated conservation practices allowed women to stay closer to home ↓ gender-based violence. Improved refugee-host relationships, market access, economy.
Food and Agriculture Organization of the United Nations, 2020; DR Congo, the provinces of Ituri and Haut-Uélé [47]	Empower South Sudanese refugees through income-generating and agricultural activities	Participants provided tools, seeds; training on agricultural, nutrition, healthy living practices. Project used cash transfers to facilitate access to goods, improve livelihoods.	2000 South Sudanese refugee households, 1000 host households; ~15,000 people. 545 refugee households also provided goats to ↑ access to animal protein.	Providing cash to rural peoples, refugees allows them to meet needs while waiting for harvests, diversify livelihoods, invest in school for children, healthcare, and financing for small business ventures.
Ghattas H, 2019; Lebanon, refugee camps [48]	Establish community kitchens (CKs) as social enterprises-improve mental health, income, food security, women’s empowerment; link with school nutrition to improve kids’ diets, school attendance, performance.	Quasi-experimental, convenience sampling. 2 CKs with 1-wk training: hands-on food safety, hygiene, nutrition education, entrepreneurship. Intervention schools: subsidized healthy food sold at school, nutrition education. Control schools: nutrition education. Evals with teachers, parents, children.	Community kitchen: 51 women recruited, 33 completed the study. School program: of 847 children 5–15 years attending intervention schools, 714 participated over 2 years.	Participatory approach = compatible work, home schedules for women. 90% of intervention and 95% of control school parents responded positively. Education sessions well attended by children, not parents. Children enjoyed snacks. Food security results not presented.
Gichunge C, 2014; Australia, East Queensland [49]	Examine gardening as part of the food environment of African refugees.	Qualitative study using in-depth interviews and a questionnaire on socio-demographics. Resettled African refugees who engaged in home and community gardening and spoke English or Swahili were recruited using purposive sampling.	13 gardeners (85% female) were interviewed. 3 from South Sudan, 1 from the DR Congo, and 9 from Burundi.	3 themes: food provision-gardens ↑ access to fresh/traditional food, saved money; health improvement-gardens helped people stay active, relieve stress, ↑ self-efficacy; food environment barriers-cost, small plots, knowledge of new climate.
Giordano, 2017; Jordan, Amman, Irbid, Marfraq, and Zarqa [50]	Review model chosen to deliver cash, identify themes of change for recipients. Common Cash Facility (CCF): platform for delivering cash, provides orgs direct, equal access to common financial service provider, payment facility.	By 2016, CCF delivered >90% of cash to refugees outside camps in Jordan. Eval: efficiency, effectiveness, relevance, coverage, accountability, innovation using data from post-distribution monitoring surveys on usage patterns, effectiveness of cash, recipient satisfaction.	Unclear.	Compared to nonrecipients, recipients: ate ↑ meals/day, fruit, eggs, meat; more diverse diets; ↓ coping strategies; ↑ income, ↑ assets, ↑ expenditures. 62–73% of households ate 2 meals the previous day, 5–10% ate 1. >50% reported the most important effect of cash assistance was eating better.
Goh J, 2017; Germany, Munich [51]	Use unconditional cash transfers to ↑ knowledge of refugee spending patterns to help aid orgs create more effective programs.	Distributed €60 to each social welfare participant to spend without limitations over ten days. Participants were divided into 3 monthly income levels: <€275, €275–€400, and >€400.	30 participants of diverse demographic backgrounds	3 largest spending categories: 40% clothes/shoes, 22% food, 9% gifts. Spending on food even across all levels. Most participants felt they had little control over their lives. They appreciated independence in what they wore and ate.
Gold A, 2014; USA, North Dakota (Fargo) [52]	Evaluate a food safety map as an educational method with English language learners.	Adult primary food preparers randomly assigned to 1. Discussion map (tailored to oral culture learners): principles of food safety, 2 h session. 2. Cooking: two 2 h classes, basic cooking skills, food safety. 3. No education. Participants in map and cooking classes received a food safety kit, questionnaire.	78 individuals began the study while 73 completed the study.	88% learned cooking skills from mothers, 36% from grandmothers, 30% from books, 16% from sister, 8% from other family members. >half cooked for children, 26% for seniors. Food safety questions answered more correctly by cooking, discussion map classes than control group.
Gunnell S, 2015; USA, Utah [53]	Evaluate if Supplemental Nutrition Assistance Program (SNAP)-Ed in English as a Second Language (ESL) classes at worksite-training reached eligible population; to pilot feasibility of food receipts to evaluate purchasing before/after classes.	1-h nutrition lessons in English for 12 weeks; mandatory training as part of work. Lessons based on 2005 USDA Dietary Guidelines using objectives of SNAP-Ed for adults, youth. Topics: food safety, food groups, common acculturation challenges of packaged/processed foods, budgeting, shopping, menu planning.	98 recently resettled refugee participants. 67% completed >10 nutrition education lessons. 17 finished the work-site training program before study completion. Eligible receipts were collected from 59 participants.	Receipts identified food purchased by 25 participants 1 week prior to nutrition lessons, 49 the first 3 weeks, 18 the last 3 weeks, two 1 week after lessons completed. 93% of receipts reflected use of SNAP funds, 15% Women, Infants and Children funds. 92% supermarkets, 59% ethnic stores.
Hartwig KA, 2016; USA, Minnesota [54]	Present a mixed method eval of a gardening project hosted by churches serving Karen and Bhutanese refugees.	Mixed methods. Survey examined food behaviors, hunger, depression, gardening experience pre- and post-season, participation in food subsidy programs. Post-survey, focus groups, interviews with church volunteers.	Out of 19 churches (>1200 refugee/immigrant families), 8 church gardens purposefully sampled based on years of participation, number of gardeners, languages. 6 focus groups: 3–10 people each (48 total). 64% of gardeners completed both surveys.	Barrier: transportation. Pre-season, 64% ate fruit/veg daily vs. 78% post. 59% ate >1 veg type/day pre-season vs. 67% post. Due to lack of response pre-season, food security questions modified post. 4% indicated no food in house due to lack of resources, some went to bed hungry. 86% participated in ≥1 food subsidy programs. 92% ↓ spending in garden season.
Hashmi A, 2019; Thailand, Mae La refugee camp [55]	Create, pilot educational materials for home-based counseling of refugee mothers along the Thailand–Myanmar border to improve infant feeding and water, sanitation, and hygiene (WASH) behaviors.	Home-based, 1-on-1 counseling for mothers with 2-months old healthy term infants; monthly visits from 3–8 months = counseling, flipbook in basic English, photos on WASH, exclusive breastfeeding for infants <6 months, local food for complementary feeding of infants >6 months. Infant feeding followed WHO recommendations.	34 mothers with infants, 59% participated in the longitudinal cohort. A total of 132 household visits were conducted with a median of 7/household.	Exclusive breastfeeding: 42% at 3 months, 65% at 5 months. Handwashing: 94% at baseline, 100% at 6/9-months. Infants at 6 months fed inadequately, 5% adequate diet diversity, 10% appropriate amounts, 0% minimum acceptable diet; ↑ to 90%, 100%, 90%, by 9 months. Sanitation, safe disposal of infant stool: 16% at 6 months, 100% at 9 months.
Hidrobo M, 2014; Ecuador, provinces of Carchi and Sucumbíos [56]	Compare impact, cost-effectiveness of cash, food vouchers, food transfers on quantity/quality of food consumed. Aimed to influence behavior change, ↑ knowledge	Randomized design. Curriculum for families, pregnant and lactating women, children 0–24 months. Transfers if attendance at monthly training. Posters, flyers: food groups, daily nutritional requirements, sanitation, food preparation, eating a variety and foods that prevent iron, vit A, calcium, iodine deficiencies.	2087 households had complete food consumption data at baseline and follow-up.	All 3 modalities ↑ quantity and quality of food. Transfers = ↑ calories, vouchers ↑ diet diversity. 99% got entire transfers, 88% on time. All 3 modalities: similar nutrition gains; sig ↑ Food Consumption Score (FCS), vouchers and food ↓ % of households with poor to borderline FCS. Cash less likely than controls to borrow money. Cash $42.99/transfer, vouchers $43.27/transfer, food $58.22/transfer. Cash = least costs (e.g., travel).
Hoddinott J, 2020; Bangladesh, refugee camps [57]	Examine associations between electronic food vouchers (e-voucher) and food rations on nutritional status of Rohingya children in Bangladeshi refugee camps.	2-stage clustered random sampling. Households assigned General Food Distribution (GFD): rice, lentils, micronutrient fortified cooking oil. WHO standards: linear growth-length/height-for-age z scores (HAZ) determine stunting, thinness-weight-for-height z scores (WHZ) determine wasting, weight-for-age z scores (WAZ), mid-upper arm circumference (MUAC).	2089 Rohingya refugee households including 523 children 6–23 months. 362 children lived in households that received food rations, 161 e-voucher. 62% of households received GFD, 34% e-vouchers. 4% that received both were excluded.	36% of children in GFD households were stunted, vs. 27% of children in e-voucher households. Wasting measures comparable across groups. E-vouchers: increase in HAZ, not stunting. No associations with weight (WHZ), acute undernutrition, WAZ, or MUAC.
Ibrahim N, 2019; Lebanon, North of Lebanon and Bekaa regions [58]	Explore impact of Community Kitchens (CKs) on food security of CK workers (CWs) and Syrian refugee (SR) families.	Exploratory qualitative descriptive approach. Purposeful and geographical variation used to recruit 4 CKs in 4 areas. CKs provided both groups with food pots on regular basis.	CWs: Lebanese or Syrian women 18–65 years, involved in local CKs ≥6 months. SRs: women of childbearing age with ≥1 child, living in an Informal Tented Settlement, received or receiving hot pots from local CK ≥6 months. 8 focus groups: 4 with CWs, 4 with SRs. 15 CWs, 49 SRs.	CKs had positive impact on food security, financial, personal, psychological, societal aspects of lives. Food pots ↓ spending, met food needs. 80% of SRs = severe food insecurity vs. 40% CWs. Some SRs: choosing families for CKs not transparent/fair. CKs: ↑ variety, amount of food ↑ nutrition, health, peace of mind. CWs: financial independence empowering.
Inglis K, 2014; Turkey, refugee camps [59]	Envisioned as efficient, innovative to let families choose/purchase diverse, nutritious food with e-Food Card.	Household assistance on e- cards bi-monthly with balance at end of month returned; used in camps, nearby centres.	21 camps, over 217,000 beneficiaries in 45,000 households; 58 shops. Most families have children <5 y of age.	>90% prefer e-cards to hot meals. >70% savings vs. hot meals, eliminated waste at distributions. Challenges: ↑ prices in shops, drought.
Karama Organization, 2015; Palestine, Deheishe refugee camp [60]	Improve refugee food security, ↓ dependency on aid, empower women to ↓ stress, ↑ physical/mental health.	Gardens = 7 tubes with soil, water system, net to cover plants, create shade. In winter, plastic converts to a green house. Participants provided tools to foster initiative, creativity, ↑ self-esteem.	15 women	Fresh veg spared limited budgets. Women felt empowered contributing to family needs, ↑ self-esteem, relieved stress, ↑ quality of life. Green spaces ↑ camp environment.
Mannion CA, 2014; Canada, Calgary Alberta [61]	Assess acceptability of a nutrition resource developed to help Sudanese refugee women purchase healthy foods, navigate grocery stores.	Grounded theory analysis. Market Guide: shopping resource to aid Sudanese refugees with food choices; encourages foods rich in iron, calcium, vit D; discourages high fat, low nutrient dense. Booklet: washable, purse-sized, nutrient-dense foods, serving sizes (Canada Food Guide), grocery store map, traditional recipes. Purposive sampling for focus group, grocery store visit.	Sudanese adult women in Canada <1 y. Of 20 women invited, 8 participated in focus group, 4 also attended grocery store visit. Interviews with 2 Sudanese Canadian intake workers, a public health nurse, center’s current medical director.	Market Guide not well received. Barriers: language, unknown foods/stores, limited knowledge. Mothers’ certainty they were doing well ↓ based on ability to feed family, if children asked for western food. Often chose traditional over unfamiliar food, had ingredients shipped. Families learned from relatives, friends, community; children from school, friends.
McElrone M, 2020; USA, mid-sized cities in Southeastern region [62]	Promote healthful cooking skill development, enhance family mealtime, ↑ physical activity through reciprocal role and behavioral modeling in Sub-Saharan Africans.	Community-based cultural adaptation of iCook 4-H: out-of-school child obesity prevention; Social Cognitive Theory; 8-session cooking curriculum-diet acculturation barriers to food security. Recruitment: local refugee programs, snowballing. After baseline, dyads randomly assigned to treatment (2-months pilot), controls.	10 youth/mother dyads (5 treatment, 5 control) with youth 8–12 years and mothers ≥18 years. Burundian, Congolese refugee families.	Process eval: positive feedback. Treatment youth ↑ cooking skills, cooking self-efficacy, eating, setting healthful goals together as a family; ↓ in playing together. Treatment adults ↑ cooking, eating, playing together, kitchen proficiency, food security.
Millican J, 2019; Iraq, Kurdistan, Domiz camp [63]	Illustrate benefits of gardening, need for sustained inclusion in camp design.	Mixed methods: ground canvassing to assess the current state of urban agriculture/gardening in camp, focus-groups, key informant interviews with families and individual refugees, and data about participants’ gardens and whether they had a garden before.	Focus groups: 1 male, 1 female. Key informant interviews: 10 families, 16 individual refugees from the 2017 garden competition, and data on 139 participants.	>50% said gardens important for mental health, wellbeing. Growing food important, relax, relieve stress; supplement income, feel happier; share/trade seeds. Women: ↑ social network, where kids play, find fresh veg. Motivators: ↑ taste, ‘clean water’. Challenges: ↓ space, water (recycle greywater).
Mochizuki Y, 2017; Uganda, Adjumani District [64]	Examine livelihood strategies of South Sudanese refugees.	Semi-structured interviews with Dinka people. Refugees given food rations, 25 m × 25 m plot of land for food. Rations: sorghum, unpopular with refugees from South Sudan, still in grain form, often pay Ugandans to produce flour.	25 households, mostly women	Most grew food common in South Sudan; 5 households grew sorghum from rations, sold to host community; 4 bred livestock; 13 grew: maize, okra, pumpkin, sorghum, chard, onion, sesame, tomato, peanut, cabbage.
Ngwenyi E, 2019; Cameroon, Far North, East, and Adamaoua regions [65]	Prevent malnutrition in children, pregnant and lactating women; ensure nutrition of nonmalnourished children, already malnourished = same supplement in regular moderately acute malnutrition (MAM) programs. Target refugees, internally displaced, hosts.	Super Cereal Plus to children 6–24 months to prevent MAM, 6–59 months to treat MAM. Social, behaviour change: infant and young child feeding (IYCF); water, sanitation, and hygiene; cooking locally available nutritious foods. Other services: e.g., immunisation, deworming, malaria prevention, supplementation, family planning, capacity-building of health workers.	Beneficiaries of supplementary feeding ↑ from 24,000 in 2015 to ~100,000 in 2016/2017. 70% of eligible received SNF, 90% participated in 66% of distributions. 1624 children 24–59 months referred to prevention program after recovery from severe acute malnutrition.	A monthly surveillance system is now in place to detect malnutrition early.
Oka R, 2011; Kenya, Kakuma Refugee Camp [66]	To exemplify the need for informal economies in refugee camps to sustain them as “urban” settlements or “refugee camp towns”	Semi-structured interviews, observation of trader-refugee-relief agency interactions. Questions covered role of informal economy in sustaining life at Kakuma, importance for traders, refugees, relief agencies.	78 traders (wholesalers, retailers); 179 refugees; 38 relief workers (UN Agencies, others).	From 2008 to 2011, food retail shops ↑ from 7 to 56, wholesalers from 4 to 8. Quantity, quality of goods, services from aid agencies affected by donor funding, supply chain, distribution = chronic malnutrition, low-quality shelter, education, training. Frequent shortages due to droughts, crop failures, budgets, transport costs. When WFP staff not present, given less. Amount of food not enough, children hungry, women went without. Trading/purchasing = dignity, power, normalcy.
Pavanello S, 2018; Greece (mainland and islands) [67]	Meet basic needs, housing, services to refugees, asylum seekers.	Emergency Support to Integration and Accommodation program delivered multipurpose cash assistance. Eval: primary, secondary data through key informant interviews, monitoring and eval data on cash program, other relevant studies, focus groups with beneficiaries of multipurpose cash assistance.	Beneficiaries of the Greece Cash Alliance program totaled 39,233, including 6000 refugees the majority of others asylum seekers. 43% were Syrians, 20% Iraqis, 19% Afghans, rest from Iran, Palestine, Pakistan, Kuwait, others. 44% located in Athens, 26% on the islands, 17% in Central Macedonia.	Cash: ↑ dignity, sense of safety, well-being; allowed preferred foods; ↓ intra-household tension. Rations/catered meals described as inedible, wasted. Women liked cooking, cleaning-alleviated boredom. Majority spent cash on food, amount not enough. Coping strategies: ↓ adult food quality/quantity; ↓ meat, milk, baby formula; borrow. ↓ information on expenditures, food security.
Qleibo E, 2013; Palestine, Gaza [68]	Cash vouchers targeted nonrefugees so not reported here. Rabbit raising program targeted refugees, nonrefugees to ↑ consumption of fresh meat, provide something to sell at local markets.	Program targeted those in need, female-headed households; each received 4 female and 1 male rabbit, cages, 200 kg fodder, a veterinary kit, training. Survey administered 4 months after receiving rabbits, 2-years profitability analysis.	286 Gazan households	98% ate, sold, donated meat. Rabbits tripled in 1–4 months. 71% ↓ debt, 52% avoided crisis sales of assets. Sustainability high: strong sense of ownership, knowledge, skills; ↓ maintenance, operational costs; commitment by partners. 2 years after implementation, 50% still operating, return ↑ >2x.
Sebuliba H, 2014; Jordan, Amman, Mafraq, Irbid and Zarqa regions including Za’atari and Azraq camps [69]	Introduce Targeted Supplementary Feeding Programme (TSFP) to treat moderately acute malnourished (MAM) Syrian children and women in camps, urban communities; ensure access to age-appropriate food.	To recruit for TSFP, Mid Upper Arm Circumference (MUAC) used to screen children under 5 years, pregnant and lactating women (PLW), girls. Those diagnosed with MAM provided SuperCereal Plus. Follow-up survey. Blanket complementary food aid (SuperCereal Plus) provided monthly to all children 6–23 months in camps.	Za’atari camp: 223 (168 children, 55 PLW, girls). Local community: 215 (79 children, 140 PLW, girls) (numbers reported as published). Blanket assistance reached an additional 8258 children <5 years in Za’atari, 456 in Azraq.	Za’atari: 68% cured, 23% defaulted, 9% transferred to outpatient care. Local community: 71% cured, 22% defaulted, 7% nonresponders. Improved acute malnutrition, GAM in Za’atri, local community. Micronutrient deficiencies persist. Prevalence of anemia: 50% in children <5 years, 64% in <2 years. Anaemia 45% in girls, women of reproductive age.
Smock L, 2020; USA, Massachusetts [70]	Improve growth parameters, anemia in low-income pregnant and breastfeeding women, children <5 years.	Special Supplemental Nutrition for Women, Infants, and Children (WIC): healthy food, nutrition education, breastfeeding support. Children divided by 2–4 visits or ≥5 visits. Measures: hemoglobin; height, weight, percentiles, z-scores for BMI, weight-for-height z-scores (acute undernutrition), height-for-age z-scores (chronic undernutrition); used 2000 CDC growth charts.	62% of refugee children <5 years who arrived in Massachusetts from 1998–2010 participated in WIC. 779 children with at least 2 WIC visits included in analyses.	Of 73 children with low weight-for-age at 1st visit, 79% recovered by last visit; of 78 with low height-for-age, 77% recovered; of 36 with low weight-for-height, 78% recovered; of 191 with anemia, 80% recovered. Females averaged 3.5 visits until recovery, males 4.3 visits. Those who remain in WIC may recover better than children with fewer WIC visits.
Stuetz W, 2016; Thailand, the Western region [71]	Evaluate impact of dietary changes on micronutrient status in each trimester of pregnancy.	Micronutrient fortified flour (MFF) as supplementary food ration to all in Maela camp, additional oil ration for pregnant and post-partum women.	533 out of 764 women (70%) participated in first survey, 515 out of 745 (69%) in second survey.	MFF ↑ daily intakes, particularly vit A, B-vits, ascorbic acid, zinc, iron; supplementary oil ration ↑ tocopherol intakes. Mean hemoglobin, high prevalence of anemia (60%), iron deficiency (39%) in 3rd trimester constant.
Sub V, 2018; Lebanon, suburbs of Beirut [72]	Address food security, economic resilience of Syrian refugees and vulnerable Lebanese host communities.	Urban gardening: horizontal, vertical, composting kits; community-based approach. Training workshops: maintenance, fertilizer, pest control, hands-on session to plant kits, raise seedlings. 4 participants received extra training to support others, monitor progress. Eval: descriptive quantitative research design.	Intervention: 73 households. Eval: 41. 71% Syrian, remainder Lebanese. All except 1 were female.	Horizontal kits in 21 households, vertical in 4, combination in 16. 24% spent less on food, 71% ↑ fruit/veg intake, 50% covered 20% of meals with garden produce. 68% produced 5–9 crops, 9% <5 crops. No success selling produce: expenses (packaging) ≥profits. <30% satisfied due to limited production, expectation to profit selling surplus. 76% planned to continue gardening, would recommend to others.
Tomkins M, 2019; Iraq, Kurdistan, Domiz camp [73]	Home gardening and tree planting	Mixed methods: Ground canvassing to assess current state of urban agriculture, gardening in Domiz Camp, focus-groups divided by gender, key informant interviews with families and refugees from 2017 garden competition, survey data from competition participants about what gardens contained and whether they gardened before. Tools, seeds, trees provided.	2 focus groups, key informant interviews: *n* = 26, and 139 surveys.	Key themes: therapeutic value of gardening; use of space for health, privacy, community; use of gardening as release from frustrations, boredom. Food gardening widely evident but not dominant in camps. Food production ranged from one family growing a single crop for cash to micro-allotment gardens of multiple veg.
Trapp M, 2010; USA, Midwest, South, and West regions [74]	Food and Nutrition Outreach (FNO) program to promote communication of culturally relevant nutrition information to newcomers to consider how social meanings, socioeconomic processes facilitate changes in food practices.	FNO: visual nutrition flipchart, training manual, poster, handouts on malnutrition, healthy eating, shopping, healthy weight, breast milk, pregnancy nutrition, exercise, nutrition labels; in 15 languages. 16 training sessions: cultural competency, nutrition, links to disease, behavioral change, nutrition outreach tools, action planning. Eval survey with trainees. Focus groups with refugees on outcomes: nutrition knowledge, dietary change, healthy eating.	Training sessions reached 200 orgs, 453 service providers. Eval surveys completed by 89 trainees. 6 focus groups conducted with 45 participants (Karen, Burundian, Congolese, Ethiopian, Burmese, West African, Hmong youth and adult refugees).	75% of trainers conducted nutrition outreach after attending training session. 75% used FNO flipchart/handouts. Some refugees and service providers took steps to ↓ fat, sugar intake. Positive changes: food decisions of agency (e.g., healthier options, milk), behaviour change of parents at day care (healthier foods), nutrition education at schools-help children use nutrition facts tables, pass knowledge to parents.
Volpato G, 2013; Western Sahara, northern Mauritania, and Tindouf [75]	Assess how role of ethnobiological knowledge and practices for refugees’ agency, through use and commodification of desert truffles, affects Sahrawi refugees of Western Sahara.	Semi-structured and retrospective interviews; “walk in the woods” approach in northern liberated territories with knowledgeable truffle harvesters (nomads and refugees).	28 semi-structured interviews, 8 retrospective interviews. The “walk in the woods” approach with 4 informants.	Truffles: delicacy, complementary, medicinal, emergency food. Resources for harvesting: knowledge (traditions taught by older refugees), access to territories, capital for commercial harvesting. Commodification generates income, recovers traditional knowledge; ↑ harvesting, competition; unsustainable.
Volpato G, 2014; Western Sahara [76]	Understand Sahrawi refugees’ agency for recovery, adaptation of traditional subsistence, other material, cultural practices. Focused on Ch. 2: camel husbandry-camels, food security (Ch. 7 on truffles examined in published article (Volpato, 2013).	Mixed methods analysed Sahrawi refugees’ recovery, adaptation of traditional practices in desert environment including camel husbandry. Surveys, interviews, focus groups, observation, ‘walk-in-the-woods’ approach, free-listings, ethnobiological voucher specimen collection.	Open interviews: 44 camel owners, 30 nomads. Semi-structured interviews: 36 refugee and nomadic camel owners (from open interviews). Focus groups: 5 with refugee camel owners and older refugees.	Camel husbandry = traditional staple foods where agriculture barely possible; hunting, gathering limited. ↑ success if own vehicle, GPS, satellite phones, new wells, pumps, tanker trucks. Sahrawi camp: positive impact on regional economies acting as hubs to sell animals. Revitalised cultural significance of camel as symbol of ethnic identity.
Wilson A, 2012; Australia, Victoria [77]	Determine standard meal weight using evidence-based nutrition principles, method to convert food collected by SecondBite into correctly defined standard, nutritionally acceptable meals, meet 30% nutritional needs of avg adult.	Cross-sectional. Used food collection database over 3 months. Observation, probing on social process of food collection, management, distribution. Used Australian Guide to Healthy Eating (AGHE) manual to develop definition of standard meal, assess nutritional quality at 2 charities; calculated using FoodWorks software.	Adults 19–60 years	% nutritional requirements 20:30:30:20, for breakfast, lunch, dinner, snacks. Total weight 30% of AGHE’s recommendations ~500 g, (10% breads, 26% veg, 26% fruit, 25% dairy, 9% meat and alternatives, 4% other); =30–36% energy, 60–65% protein, 64% vit C, 76% calcium, 38% iron for men, 17% iron for women.
World Food Programme, 2014; Mozambique, Maputo and 4 Districts in Gaza Province; 3 Districts in Zambezia Province; and 1 refugee camp in Nampula Province (Maratane camp) [78]	Support populations who become transiently food insecure as a result of recurring seasonal shocks to: save lives, protect livelihoods in emergencies, restore/rebuild lives, livelihoods in post-conflict, post-disaster or transition situations; strengthen capacities of countries to ↓ hunger.	General Food Distribution (GFD) to disaster-affected households, refugees at Maratane; Food for Assets (FFA) to implement activities to rehabilitate assets, maintain food security; capacity development for gov stakeholders. Eval: mixed methods. Qualitative: in-depth structured, semi-structured interviews with WFP staff, stakeholders; focus groups (by gender) with beneficiaries, nonbeneficiaries, stakeholders; direct observation.	8000 refugees in Maratane (only location where humanitarian aid offered, interviews for refugee status conducted); 2805 asylum seekers, 718 refugees on outskirts. Eval: document review, session with CO, stakeholders’ workshop, 2 debriefs to present preliminary findings, 110 stakeholder interviews, 33 focus groups with 205 women, 185 men.	Refugees in camp can travel out for work if registered. Food rations appropriate since markets not fully functioning. FFA did not appropriately target most in need. National capacity for contingency planning, food security, emergency assessment ↑; sustainability still a concern. Effective, efficient supply chain management strongest asset of program, saving lives, ↑ food security, exceeding GFD targets, timely delivery. Underfunding affected FFA targets in 2012, GFD tonnage targets in 2013.
World Food Programme, 2015a; Jordan, Lebanon, Iraq, Turkey and Egypt [5]	Provide food assistance to vulnerable households whose food, nutrition security adversely affected by civil unrest in Syria, to save lives, protect livelihoods in emergencies.	Syria: Targeted General Food Distribution (GFD) as household in-kind rations, blanket in-kind supplementary feeding for children 6–59 months, vouchers for pregnant/lactating women, in-kind school-feeding. Lebanon: GFD to eligible out of camp refugees, returning Lebanese as 1-month in-kind parcels then vouchers. Turkey: GFD to all refugees in camps as vouchers. Jordan: GFD to all refugees in camps as 1-day in-kind meals then vouchers with daily bread, vouchers to all registered refugees out of camps, in-kind school feeding in camps, in-kind nutrition for refugees in and out of camps. Iraq: GFD to all refugees as in-kind in 9 camps, vouchers in 1 camp, in-kind school feeding in 2 camps. Egypt: GFD as vouchers to eligible out of camp refugees, Palestinians from Syria. Fieldwork in Jordan, Lebanon Turkey; remote collection in Egypt, Iraq, Syrian Arab Republic. Interviews, focus groups, stakeholder questionnaire.	Response scaled up quickly, assisting 4.25 M people in Syria, 2 M refugees across the region. WFP reached 88% of target in Egypt, 98% of all registered refugees in Jordan. Eval: 259 (55% women) interviews, 47 focus groups, 32 responses to stakeholder questionnaire (majority from Syria). In Jordan, Turkey, eval further considered views from refugees in and outside camps; data from host communities collected where feasible in Lebanon, Turkey.	Issues: timely baseline data, inconsistent staffing, inadequate oversight, WFPs proximity to Syrian gov. New refugees in Jordan (12%) and Lebanon (16%) had poor FCS, vs. 4% and 3% of refugees on aid. Acceptable FCS on arrival 50% in Lebanon, improved to 78–98% (all countries). Focus groups: importance of food aid-main source of income for purchasing food. Most common food coping strategy: less preferred/less expensive food. Aid ↓ coping strategies. Benefits to local economies, refugee–host relationships improved with e-vouchers. In-kind food usually on time, vouchers subject to delays. Vouchers periodically resulted in > normal market prices. WFP built complex transport/logistics network to prevent inappropriate relationships with armed groups by rotating companies, drivers, routes.
World Food Programme, 2015b; Iran, provinces of Fars, Kerman, Khorasan-e-Razavi, Markazi, West Azerbaijan, and Yazd [79]	Improve food consumption of vulnerable refugees, ↑ access to education and human capital development for refugee girls, youths.	General Food Distribution (GFD) with target approach-2 levels of household vulnerability, 2 food rations; School Feeding-take home rations to women teachers, girls in primary/secondary schools; Food for Training- take home rations to trainees. Midterm eval: secondary data, semi-structured interviews of focus groups and individual households, interviews with stakeholders, observation, internal/external debriefings.	30,000 refugees and 200 teachers.	Targeting process lacked accuracy (selection criteria), participation. Most aid through GFD, ↓ support for livelihood strategies. Targets reached despite operational limits (i.e., import constraints due to sanctions on Iran) = sig ↓ deliveries than planned. Lack of measure, unreliable indicators = difficult to analyse. Food consumption maintained or ↑ with aid for refugees in settlements.
World Food Programme, 2016a; Ethiopia, Gambella, Afar, Tigray (Shire), BenishangulGumuz (Assosa) and Somali (Dolo Ado and Jijiga) [80]	Assess previous operation’s transition period, performance of current operation to ensure informed decisions, future design strategies.	General food distribution, school feeding (SF), blanket and targeted supplementary feeding, livelihood support. Eval: lit review, observation, in-depth semi-structured interviews. >half of country’s households are food insecure as defined as per capita access to calories.	130 key informant interviews, 35 focus groups with 401 participants (207 female, 194 male).	Supplementary feeding reached vulnerable children, mothers; global acute malnutrition still ↑. Women, children collect firewood-↑ risk of gender-based violence. Food rations 89% to 95% of target, cash 89% to >100%. Household diet diversity, food consumption met targets. Food distributions fair, smaller households (women) at disadvantage. SF 44 to 79% of target, satisfactory, appreciated. School retention exceeded targets. Cash transfers: ↓ sale of food aid, ↑ choice, flexibility; somewhat ↑ food eaten, ↑ empowerment, dignity. Cash, biometrics ↓ fraud, need to sell rations.
World Food Programme, 2016b; Liberia, Maryland, Grand Gedeh, and Nimba counties [81]	FFA objective: to protect livelihoods, create assets for vulnerable host populations, refugees living in host communities. School Feeding objective: support enrolment, retention in school.	General Food Distribution (GFD), School Feeding (SF), Food for Assets (FFA) to ↑ access to markets, enhance agriculture production by providing each participant with 120 days of work. For children 6–59 months, targeted supplementary feeding (TSF) for Moderate Acute Malnutrition (MAM) in host communities, Stunting Prevention Programme in host communities and refugee camps. Eval: document reviews, key informant interviews, observations. Standardised Expanded Nutrition Surveys assessed nutrition status.	Food aid to 100,136. Eval: 370 persons interviewed during key informant interviews and 35 disaggregated focus groups (65 community members and 137 refugees in camps and host communities). SF in 3 primary schools in 3 refugee camps assisting 7694 children.	Liberia affected by Ebola, impacted program delivery, delayed repatriation. Gaps, inaccuracies found in data. Effectiveness limited-all activities except GFD suspended (↓ funding). GFD prioritized in camps, (an approach that promoted self-sufficiency, addressing vulnerable groups would have ↓ dependency on external aid). 73% planned GFD delivered maintaining nutritional status, few outcome targets met. For children 6–59 months, acute malnutrition = acceptable WHO levels, chronic malnutrition critical, prevalence of anaemia ↓ 78% to 67%; TSF not successful; SPP should not have been suspended. SF = 77% of target; aimed to ↑ enrolment by 6%/y, inconclusive. FFA activities benefited Liberians, refugees in host communities. 20–30% of households had acceptable food security, 20–40% of rations sold to purchase other food.
World Food Programme, 2016c; Rwanda, Gihembe, Mahama, Kigeme, Nyabiheke, and Kiziba camps, and Bugesera and Nkamira transit centres [82]	Meet food/nutrition needs of refugees, returnees; treat acute malnourished children 6–59 months; prevent chronic malnutrition in children 6–23 months; prevent malnutrition while ↑ adherence to drug protocols of people living with HIV on antiretroviral treatment and patients; ↑ access, quality of education/health facilities in camps.	Food aid through general food distribution (GFD) or cash-based transfers (CBT), and nutrition and school feeding (SF) programs. SF: 1 meal/school day to children at 13 primary/secondary schools, in camps or host communities. Midterm eval: mixed methods including a document review, observation, in-depth structured/semi-structured interviews with key informants, focus groups with refugees, host communities.	GFD 81,593, CBT 49,816. Preventive feeding 19,700 children <5 years, 8458 pregnant and lactating women. Curative feeding 3255 children <5 years, 1224 medical cases. SF 34,731 primary children, including 8900 in host community. Eval: 170 key informant interviews, 29 focus groups (223 women, 105 men) plus 40 random impromptu focus groups.	Implementation efficient, effective including supply chain; no funding interruptions. Commodities = good quality, maize and beans for GFD procured locally, ~10% purchased from small farmers. Distribution facilities well run. Households with poor food security remained steady, acceptable food security ↑ to 77%. Average coping strategies index scores ↓ from 11.4 to 9.7, meeting targets. Dietary diversity was below expectations in cash camps (4.24), ahead of food camps (4.07). Nutrition program ↓ global acute malnutrition in Mahama, stabilised malnutrition for pregnant women, children. SF was a significant, well-managed activity. Few livelihood opportunities; plans for grinding mill, gardening, rabbit production in place.
World Food Programme, 2018a; Kenya, Kakuma camp, Kalobeyei settlement, and Dadaab camp [83]	Cash Based Transfers (CBT) scaled up to ↑ cost effectiveness of food aid in Kenyan Refugee Operations to develop a model that determines effective, efficient mix between food aid, CBT.	CBTs, called *Bamba Chakula* (Swahili for ‘get your food’)—introduced to all registered refugees in camps in response to ↓ dietary diversity, reselling in-kind aid at a loss. CBTs: monthly e-vouchers via SIM cards to buy food through contracted traders. CBTs substituted cereal rations, began by replacing 10%, ↑ to 30–50%. Eval: mixed methods including quantitative surveys, in-depth interviews, focus groups, key informant interviews. Data was gender disaggregated. Comparison due to lack of control group.	Food, CBT to all refugees (146,7682 Kakuma, 38,170 Kalobeyei, 235,2964 Dadaab). Some refugees living in camps 25 years. Quantitative surveys administered to 542 households in Kakuma, 545 households in Kalobeyei; 230 traders, 626 households from host and nonhost communities. Refugees from South Sudan 56.4%, Somalia 20%, Ethiopia 5.6%.	Coverage ↑, but below target. Rations sold as sorghum unfamiliar, disliked, to purchase other items. Traders ↑ price; delays in disbursements = credit purchases, mostly female households = loyalty, indebted to traders. Kakuma = ↓ food security, diet diversity; ↓ nutritious foods than Kalobeyei due to ↑ transfer value in Kalobeyei, ration cuts, delayed disbursements, ↓ purchasing power in Kakuma. Kalobeyei = ↑ severe hunger, asset poverty, ↓ livelihood opportunities, worse gender equality. Female households worse across indicators. Positive impacts on livelihoods, food security in host vs. distant communities. CBT more cost-efficient than transfers.
World Food Programme, 2018b; Algeria, 5 refugee camps near Tindouf [84]	Improve food consumption of most vulnerable refugees through food aid, to ↓ acute malnutrition, anaemia in children <5 years, pregnant and lactating women (PLW) through targeted nutrition interventions; to maintain enrolment/retention of children.	Nutrition components used General Food Distribution (GFD), prevention, treatment of undernutrition, anaemia among children <5 years, PLW, School Feeding (SF). Eval used mixed methods: key informant interviews, focus groups, field visits, storytelling, Photovoice.	Monthly planned GFD rations targeted 125,000. Targeted Supplementary Feeding Programme 1800 children/month, 1000–6000 PLW/month. Preventive component 13,250 children/month, prevention of anaemia 6360 PLW/month, SF 32,500 children/month. Numbers may not be accurate due to data discrepancies.	Financial limitations ↓ diversity, nutrition of food aid. Most nutrition, food security outcomes sig ↑. Prevalence of acute and chronic malnutrition ↓ in children, below emergency levels, underweight residual in women. Sig ↑ of overweight, obesity, metabolic risk contributing to double burden of undernutrition and obesity. Those who could afford it purchased food to complement rations. Photovoice = valued foods not often distributed. Satisfied with aid, request ↑ quantity/quality, regular distribution. Diets did not meet requirements for calcium, iron, niacin, vit C, vit A. Acceptable food security ↑ 77% to 93%.
World Food Programme, 2018c; Cameroon, the Far North, North, East and Adamaoua regions [85]	Improve resilience to address chronic and acute malnutrition, food insecurity, household vulnerability towards climatic hazards.	General food assistance (GFA) introduced then cash-based transfers (CBTs). Nutrition interventions treated malnutrition through targeted supplementary food to children <5 years, pregnant and lactating women with blanket supplementary feeding as a complement for children 6–23 months. Food for assets (FFA) for refugees, host populations with moderate food insecurity. Food by prescription for malnourished people living with HIV, receiving anti-retroviral therapy. School feeding implemented.	GFA 1,268,998 (104% of target), nutrition activities 1,879,003 (86% of target), school feeding 91,728 (25% of target), FFA 397,648 (55% of target), Food by prescription 3819 (89% of target). Refugees from Chad, Nigeria, Central African Republic.	CBTs, shift from nutrition treatment to prevention = positive. ↑ efficiency with CBTs, mobile vulnerability analysis, mapping for data collection in areas with restricted access. Funding shortfalls, delays = ↓ rations, temporary suspensions of distributions, cessation of school meals. Sustainability of FFA activities limited. Food insecurity in Cameroon ↑ to 10%. 32% of children <5 years chronically malnourished, 13% severely stunted. Chronic and acute malnutrition high in N., E. Cameroon, improved in the Far N., N., Adamaoua when WFP provided support. Malnutrition deteriorated in E. With ↑ refugees from Central African Republic.
Wtsadik M, 2011; Ethiopia, Shimbelba, Awbarre, and Kebribeyah refugee camps [86]	↑ availability of veg, eggs at household level, thereby ↑ micronutrient status of vulnerable refugees.	Multi-storey gardens (MSG), poultry provided to 3 camps. Oil cans filled with rocks in 50 kg cereal bags with holes in top, sides. Seeds planted on top, thinned out, inserted in sides. Required 5 L water 2x/day-recommended greywater. Each household encouraged to build 5 MSGs, provided 3 poultry (1 male, 2 females). Targeted family members with anaemia or malnutrition, large female-headed families, people with HIV/AIDS. Eval: questionnaires on veg consumption, veg sold, % rations sold to buy veg, water use, egg consumption.	167 households in each of 3 camps. Eval: 50 households (random selection). Focus groups: 15–20 households (random selection) not included in household survey and 5 households who were not one of the 167 beneficiary households.	Compared to backyard gardens, MSGs needed ↓ water, veg grew faster, 2 harvests possible. Refugees acquired new skills, diverse meals, shared produce, less likely to sell rations for veg. At eval, chickens too small to lay eggs in 2 camps, but in 1 camp, 35% of participants ate ~7.5 eggs/week. Project well accepted, requested by nonpilot households; allowed refugees to choose what to plant/eat, gave sense of dignity, well-being. Some refugees trying to duplicate MSG on their own. Poultry not recommended: chickens ate produce, ↑ cost, time.

↑ = increase; ↓ = decrease.

**Table 2 nutrients-14-00522-t002:** Themes of eligible studies.

First Author, Year	Location	Target Population	Intervention Type	Considers Gender	Food Security Measure
World Food Programme, 2018b [84]	Algeria: Refugee camps/informal settlements	Refugees only	Mixed cash, vouchers, food transfers	No	Dietary Diversity Score, Food Consumption Score, Coping Strategies Index
Wilson A, 2012 [77]	Australia: Destination country	Refugees and host communities	Nutrition education programming	No	No
Gichunge C, 2014 [49]	Australia: Destination country	Refugees only	Urban agriculture, animals, foraging	No	No
Hoddinott J, 2020 [57]	Bangladesh: Refugee camps/informal settlements	Refugees only	Mixed cash, vouchers, food transfers	Yes	No
Ngwenyi E, 2019 [65]	Cameroon: Inside and outside camps	Refugees and host communities	IYCF * and pregnancy	Yes	No
World Food Programme, 2018c [85]	Cameroon: Inside and outside camps	Refugees and host communities	Mixed cash, vouchers, food transfers	Yes	Dietary Diversity Score, Food Consumption Score, Coping Strategies Index
Mannion CA, 2014 [61]	Canada: Destination country	Refugees only	Nutrition education programming	Yes	No
Food and Agriculture Organization of the United Nations, 2020 [47]	DR Congo: Outside camps	Refugees and host communities	Urban agriculture, animals, foraging	Yes	No
Hidrobo M, 2014 [56]	Ecuador: Outside camps	Refugees and host communities	Mixed cash, vouchers, food transfers	Yes	Food Consumption Score
World Food Programme, 2016a [80]	Ethiopia: Inside and outside camps	Refugees only	Mixed cash, vouchers, food transfers	Yes	Per capita access to calories
WTsadik M, 2011 [86]	Ethiopia: Refugee camps/informal settlements	Refugees only	Urban agriculture, animals, foraging	Yes	No
Goh J, 2017 [51]	Germany: Destination country	Refugees only	Cash	No	No
Dunlop K, 2018 [23]	Greece: Inside and outside camps	Refugees only	Cash	No	Questions on challenges in accessing shops/markets; travel time to nearest market/shop; travel costs; food/nonfood items in the last 2 months.
Pavanello S, 2018 [67]	Greece: Inside and outside camps	Refugees only	Cash	Yes	No
World Food Programme, 2015b [79]	Iran: Outside camps	Refugees only	Mixed cash, vouchers, food transfers	Yes	Food Consumption Score and Diet Diversity Score
Tomkins M, 2019 [73]	Iraq: Refugee camps/informal settlements	Refugees only	Urban agriculture, animals, foraging	No	No
Millican J, 2019 [63]	Iraq: Refugee camps/informal settlements	Refugees only	Urban agriculture, animals, foraging	No	No
Giordano N, 2017 [50]	Jordan: Outside camps	Refugees only	Cash	No	Questions on # of meals, diet diversity, food frequency, coping strategies.
Fander G, 2014 [44]	Jordan: Inside and outside camps	Refugees and host communities	IYCF * and pregnancy	Yes	No
Alsamman S, 2014 [34]	Jordan: Refugee camps/informal settlements	Refugees only	IYCF * and pregnancy	Yes	No
Sebuliba H, 2014 [69]	Jordan: Inside and outside camps	Refugees only	IYCF * and pregnancy	Yes	No
Boston Consulting Group, 2017 [39]	Jordan and Lebanon: Outside camps	Refugees only	Mixed cash, vouchers, food transfers	No	Consolidated Approach for Reporting Indicators
Abu Hamad B, 2017 [32]	Jordan: Outside camps	Refugees only	Mixed cash, vouchers, food transfers	No	Food Consumption Score
World Food Programme, 2015a [5]	Jordan: Inside and outside camps	Refugees only	Mixed cash, vouchers, food transfers	No	Food Consumption Scores and the Coping Strategy Index
World Food Programme, 2018a [83]	Kenya: Refugee camps/informal settlements	Refugees only	Cash	No	Household Dietary Diversity Score, Food Consumption Score, Coping Strategies Index
Betts A, 2020 [37]	Kenya: Inside and outside camps	Refugees only	Mixed cash, vouchers, food transfers	No	Household Food Insecurity Access Prevalence; Dietary Diversity Score; Food Consumption Score
Oka R, 2011 [66]	Kenya: Refugee camps/informal settlements	Refugees only	Informal economy/trading	No	No
Battistin F, 2018 [36]	Lebanon: Outside camps	Refugees only	Cash	No	Food Consumption Score, Household Weekly Dietary Diversity Score, Household Daily Average Dietary Diversity Score
Ibrahim, N., 2019 [58]	Lebanon: Refugee camps/informal settlements	Refugees and host communities	Community kitchen	Yes	Questions on types, amount, and variety of food, nutritional value, for chronic conditions, preference, culture, finances, what happens when food runs out, supplemental food.
Ghattas H, 2019 [48]	Lebanon: Refugee camps/informal settlements	Refugees only	Community kitchen/School-based nutrition	Yes	Arab Family Food Security Scale
Aste N, 2017 [35]	Lebanon: Outside camps	Refugees only	Food safety and energy	No	No
El Harake MD, 2018 [43]	Lebanon: Refugee camps/informal settlements	Refugees only	School-based nutrition	Yes	Arabic-translated, locally validated version of the Household Food Insecurity Access Scale
Dehnavi S, 2019 [41]	Lebanon: Outside camps	Refugees and host communities	Urban agriculture, animals, foraging	No	No
Food and Agriculture Organization of the United Nations, 2016 [45]	Lebanon: Inside and outside camps	Refugees and host communities	Urban agriculture, animals, foraging	Yes	No
Sub V, 2018 [72]	Lebanon: Outside camps	Refugees and host communities	Urban agriculture, animals, foraging	No	6-item short form of 18-item Food Security Measurement Module by the United States Department of Agriculture
World Food Programme, 2016b [81]	Liberia: Inside and outside camps	Refugees and host communities	Mixed cash, vouchers, food transfers	No	Food Consumption Score
World Food Programme, 2014 [78]	Mozambique: Inside and outside camps	Refugees and host communities	Mixed cash, vouchers, food transfers	No	Emergency Food Security Assessment, Comprehensive Food Security and Vulnerability Analysis
Qleibo E, 2013 [68]	Palestine (Gaza): Outside camps	Refugees and host communities	Urban agriculture, animals, foraging	Yes	No
Karama Organization, 2015 [60]	Palestine: Refugee camps/informal settlements	Refugees only	Urban agriculture, animals, foraging	Yes	No
Alloush M, 2017 [33]	Rwanda: Refugee camps/informal settlements	Refugees only	Mixed cash, vouchers, food transfers	No	One question: “In last 7 days, have there been times when household did not have enough or money to buy food?”
World Food Programme, 2016c [82]	Rwanda: Inside and outside camps	Refugees and host communities	Mixed cash, vouchers, food transfers	Yes	Food Consumption Score
de Bruin N, 2019 [40]	Tanzania: Refugee camps/informal settlements	Refugees and host communities	Mixed cash, vouchers, food transfers	No	No
Hashmi A, 2019 [55]	Thailand: Refugee camps/informal settlements	Refugees only	IYCF * and pregnancy	Yes	No
Stuetz W, 2016 [71]	Thailand: Refugee camps/informal settlements	Refugees only	IYCF * and pregnancy	Yes	No
Inglis K, 2014 [59]	Turkey: Refugee camps/informal settlements	Refugees only	Voucher	No	No
Mochizuki Y, 2017 [64]	Uganda: Refugee camps/informal settlements	Refugees only	Urban agriculture, animals, foraging	No	No
Food and Agriculture Organization of the United Nations, 2018 [46]	Uganda: Outside camps	Refugees and host communities	Urban agriculture, animals, foraging	Yes	No
Smock L, 2020 [70]	USA: Destination country	Refugees only	IYCF * and pregnancy	Yes	No
Trapp M, 2010 [75]	USA: Destination country	Refugees only	Nutrition education programming	Yes	No
Bloom JD, 2018 [38]	USA: Destination country	Refugees and host communities	Nutrition education programming	Yes	No
Gold A, 2014 [52]	USA: Destination country	Refugees and host communities	Nutrition education programming	No	No
Gunnell S, 2015 [53]	USA: Destination country	Refugees only	Nutrition education programming	No	No
McElrone M, 2020 [62]	USA: Destination country	Refugees only	Nutrition education programming	Yes	Measured, tool not specified. Food Security Score: 26 items for adults- cooking, eating, playing together, kitchen proficiency, and food security.
Eggert LK, 2015 [42]	USA: Destination country	Refugees only	Urban agriculture, animals, foraging	No	No
Hartwig KA, 2016 [54]	USA: Destination country	Refugees and host communities	Urban agriculture, animals, foraging	No	Hunger assessed using internationally validated food security questions by the UN Food and Agriculture Organization
Volpato G, 2013 [75]	Western Sahara, Mauritania, Algeria: Inside and outside camps	Refugees only	Urban agriculture, animals, foraging	No	No
Volpato G, 2014 [76]	Western Sahara: Inside and outside camps	Refugees only	Urban agriculture, animals, foraging	No	No

* IYCF = Infant and Young Child Feeding.

## Data Availability

Data sharing not applicable to this scoping review.

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
