# Peer review of "Food Security Interventions among Refugees around the Globe: A Scoping Review"

_nutrients, 2022, doi:10.3390/nu14030522_

Round 1

Reviewer 1 Report

The manuscript fits very well in the aim of this Special Issue "Food Security, Food Intake and Eating Behaviour in Low- and Middle- Income Countries"

The manuscript is well written and the content is interesting. The introduction could be shortened and the discussion section could be expanded. In the introduction, as well as in the whole manuscript, children, elderly and sick people should also be mentioned as a risk population and not only women.

Figure 1 is not included in the manuscript, if it is listed in the supplementary material, then it should not be listed as Figure 1 in the manuscript. Figures 3 and 4 would be significantly valorized, if they were colored.

In Figure 2 «PRISMA Flow Diagram”, please add in the boxes “screening, exclusion” and “eligibility, exclusion” the most important reasons for exclusion.

Table 1 is very complex and too long. Please shorten clearly and really concentrate only on the most important results and statements, best to write in telegram style. Must be shortened in the minimum in one third to one-half. One possibility is to move the complete table in the additional material and to reduce the table in the manuscript significantly so that important information is not lost.

Please add after the gender consideration a small section/consideration (2-3 statements) about the risk population children, elderly and sick people.

In the conclusion remarks you propose further research and to use a common food security measurement. According to your experience and after you have delved deeply into the problem, what would you suggest for measurements tools or what should these tools contain most importantly? Do you have some suggestions for the readership? This would be an interesting information to add in the section conclusion or in the discussion.

The references are up to date and well chosen.

Author Response

Dear Reviewer,

We would like to sincerely thank for your valuable comments. We revised the manuscript based on the comments. The parts of the manuscript with major changes are highlighted. Following, please find our point by point responses to your comments. 

Reviewer 2 Report

I applaud the authors for their effort in addressing an increasingly important issue – food security among refugees. The amount of work involved in summarizing 57 studies is worth credits in and of itself. The introduction and methodology are clear and informative. Below I present comments that I hope may further strengthen the paper.

My main concern is that the “Discussion” section may fall short on informing readers about the effectiveness, efficiency, equity among other evaluation criteria of the food security interventions, the supposed focus of this review study. The current Discussion only offers a very broad summary (e.g., cash and food rations as types of intervention). I as a reader was expecting greater detail on how effective, efficient, equitable the interventions were. For instances, did cash reduce food insecurity more than food rations did? Did giving assistance to women versus men lead to differential food security outcomes within and across households? Which approaches had “minimal to no success” (line 133) when programs were implemented? Examples from single studies would also help clear up these questions.

Related to my main concern above is the presentation of Table 1. The authors should consider using succinct and standardized language and bullet points to make them easier to digest. Details peripheral to the delivery of key message (such as detailed information on subsamples of a study) could be synthesized or removed. The authors may further consider grouping studies by intervention type in Table 1 and Table 2 as they claimed would do (line 174).

What are the roles of FAO and WFP, respectively? What is their relationship? Is their collaboration expected? Is there any theory or evidence on the benefits of collaboration between these organizations? Even though I could imagine their collaboration could benefit the communities, the authors should still show evidence and rationale that support this hypothesis given the potential conflict of interests and organizational bureaucracy that arise from collaboration. (line 30 and 108)

The authors discussed the role of gender in food security intervention (line 70). I hope they could elaborate on their motivation to examine “gender” and how that differs from the assumption of intra-household resource pooling (i.e., assumed gender equality in within-household food sharing). Also, if gender is considered, does that mean that food security is measured at individual versus household level? Clarification and implications of such approach are needed in the text.

Minor comments are listed below.

  • “USD were pledged” by who? UN, NGO, governments? (line 114)
  • An extra “and” before JIRS? (line 157)
  • “reference lists”? (line 158)
  • “However,” should be “While”? (line 1, p.62)
  • “food security issues” like what? (line 2)
  • “two” or “three” questionnaires? (line 8)
  • What is “Intervention types across geographic locations”? (line 19) Should it be a section header or a phrase? Should it be italicized?
  • Should “Areas of refugee crisis” (line 20) and “Destination countries” (line 60) be italicized? The formatting of these section headers are confusing.
  • “This may explain” is confusing. (line 78) Does it mean “receipt of social assistance” increase food insecurity?
  • “more food insecure” or “more likely to be food insecure”? (line 83)
  • In-text citations are needed for the “Knowledge Gaps and Research Recommendations” section. (line 103)
  • “refugee” (line 119)
  • “our knowledge” (line 131)
  • “minority groups”? (line 154)

Author Response

(The authors gave the same response as above.)

Round 2

Reviewer 1 Report

The authors made a great effort and revised the manuscript very well and to my complete satisfaction. The manuscript has now gained in quality and the publication will be very interesting and attractive for the readership. Excellent work.

Reviewer 2 Report

No further comments.